# Subclinical hypothyroidism in Wales from 2000 to 2021: A descriptive cohort study based on electronic health records

**Brenda S. Bauer** [1]*, **Amaya Azcoaga-Lorenzo** [1,2], **Utkarsh Agrawal**[3], **Adeniyi Francis Fagbamigbe**[4], **Colin McCowan**[1]

1 Division of Population and Behavioural Sciences, University of St Andrews School of Medicine, St Andrews, United Kingdom, 2 Hospital Rey Juan Carlos, Instituto de Investigación Sanitaria Fundación Jimenez Diaz, Red de Investigación en Cronicidad, Atención Primaria y Promoción de la Salud (RICAPPs) ISCIII, Madrid, Spain, 3 Nuffield Department of Primary Care Health Sciences, University of Oxford, Oxford, United Kingdom, 4 Research & Evaluation Unit, Institute for Health and Wellbeing, Centre for Healthcare and Communities, Coventry University, Coventry, United Kingdom

* bsb1@st-andrews.ac.uk

## Abstract

### Background

Subclinical hypothyroidism (SCH) is a biochemical thyroid disorder characterised by elevated levels of Thyroid Stimulating Hormone (TSH) together with normal levels of thyroid hormones. Evidence on the benefits of treatment is limited, resulting in persistent controversies relating to its clinical management.

### Aim

This study describes the demographic and clinical characteristics of patients identified as having subclinical hypothyroidism in Wales between 2000 and 2021, the annual cumulative incidence during this period and the testing and treatment patterns associated with this disorder.

### Methods

We used linked electronic health records from SAIL Databank. Eligible patients were identified using a combination of diagnostic codes and Thyroid Function Test results. Descriptive analyses were then performed.

### Results

199,520 individuals (63.8% female) were identified as having SCH, 23.6% (n = 47,104) of whom received levothyroxine for treatment over the study period. The median study follow-up time was 5.75 person-years (IQR 2.65–9.65). Annual cumulative incidence was highest in 2012 at 502 cases per 100,000 people. 92.5% (n = 184,484) of the study population had TSH levels between the upper limit of normal and 10mIU/L on their first test. 61.9% (n = 5,071) of patients identified using Read v2 codes were in the treated group. 41.9% (n =

**Data Availability Statement:** The data used for this study are third-party data provided by SAIL Databank. Access to the datasets is conditional upon application and review by their Information

Governance Review Panel. Interested parties can access the datasets by following the same application process as the authors, as detailed on the SAIL Databank website (https://saildatabank.com/data/apply-to-work-with-the-data/). The authors did not have any special access privileges that other applicants would not have.

**Funding:** This study was performed as part of a PhD studentship funded by the School of Medicine, University of St Andrews. The University of St Andrews played no role in the design and conduct of the study.

**Competing interests:** The authors have declared that no competing interests exist.

19,716) of treated patients had a history of a single abnormal test result before their first prescription.

## Conclusion

In Wales, the number of incident cases of SCH has risen unevenly between 2000 and 2021. Most of the study population had mild SCH on their index test, but more than a third of the identified patients received levothyroxine after a single abnormal test result. Patients with clinically recorded diagnoses were more likely to be treated. Given the expectation of steadily increasing patient numbers, more evidence is required to support the clinical management of subclinical hypothyroidism.

## Introduction

The thyroid gland is a butterfly-shaped endocrine organ in the neck whose function is regulated by Thyroid Stimulating Hormone (TSH) from the pituitary gland. It produces the thyroid hormones thyroxine (T4) and triiodothyronine (T3), which primarily regulate bodily metabolic function. Hypothyroidism generally refers to a deficiency of these hormones but can be subdivided into overt and subclinical types. The latter, subclinical hypothyroidism (SCH), is a frequently asymptomatic condition in which the thyroid hormone levels are within normal range, but TSH is elevated [1, 2].

Reference ranges for TSH vary due to patient characteristics, particularly age, sex, race, ethnicity and pregnancy status [3, 4]. There is disagreement over what should be considered as the upper limit of normal [5–9], so these values tend to differ between laboratories, given the lack of universally applicable guidelines. However, a commonly used upper cut-off for TSH is 4.5 mIU/L [10], such that patients with measurements between 4.5 and 9.9 mIU/L are said to have mild SCH. On the other hand, TSH levels ≥10 mIU/L are classified as severe SCH [2].

Among the causes of SCH, Hashimoto's thyroiditis, an autoimmune thyroid disorder, is the most frequent. Women are more likely to develop SCH than men, regardless of age [2, 11]. Previous studies have also reported that the population prevalence of SCH ranges between 4% and 10%, depending on factors such as age and sex distribution patterns [1, 11–13]. The Colorado Thyroid Disease Prevalence Study determined that in their study population of over 25,000 patients, approximately 9% of subjects not on thyroid medication were found to have SCH [11]. The US National Health and Nutrition Examination Survey (NHANES III) reported SCH prevalence figures of 4.3% in the total population [13].

It has been reported that around 2% to 6% of SCH cases each year experience progression to overt hypothyroidism. This phenomenon occurs more commonly among female patients and those found to test positive for thyroid peroxidase antibodies, for instance, in autoimmune thyroid disease [1, 14]. As reported in some studies, the TSH levels of approximately 60% of patients identified as having mild SCH may later spontaneously revert to normal [2, 15].

The pharmacological treatment for overt and subclinical hypothyroidism is levothyroxine (LT4), a synthetic version of thyroxine. Controversy persists on the clinical management of SCH, specifically around whether to initiate treatment and, in those cases, what level of TSH to use as a threshold. The debate is due to insufficient robust evidence and conflicting study findings on the long-term benefits–and, inversely, potential harms–of treatment for this disorder [1, 10, 16–19]. For example, many studies have investigated the effects of SCH on cardiovascular disease, but while some found that levothyroxine lowered the incidence of myocardial

infarctions, atrial fibrillation and cardiovascular mortality [20], others did not [21, 22]. Similarly, Mooijaart et al. reported that in 2 randomised trials of levothyroxine treatment for patients 80 years and above, there was no significant difference in quality of life (QoL) between treatment and control groups [23]; whereas in a cohort study of 78 patients, Winther et al. found marked improvements in health-related QoL within six months of starting treatment [24]. Even so, current NICE guidelines state that treatment should only be commenced after a repeat abnormal TSH result of $\geq$10 mIU/L three months after the first [25].

The debate extends to screening for SCH, despite the frequency of no reported symptoms, due to the lack of substantiated evidence of benefits for the majority who might be diagnosed through screening programs [2, 26, 27]. Other issues that are pertinent to the management of SCH are the overuse of thyroid function tests, also known as overtesting, which can potentially increase the detection of elevated TSH [28, 29], and the overuse of levothyroxine for SCH [19, 30], with no clarity on how it affects patients in the long-term.

Our study aims were to (i) describe the demographic and clinical characteristics of patients identified as having SCH in Wales between 2000 and 2021; (ii) estimate the annual cumulative incidence of SCH and the accumulation of patients identified as having SCH during the study period; (iii) characterise thyroid function testing and levothyroxine prescribing over the study period, including the TSH thresholds used to initiate treatment for SCH.

## Methods

### Study design

This is a retrospective, population-based cohort study using the anonymised, linked electronic health records (EHR) for the population living in Wales between 1 January 2000 and 31 December 2021. These were individuals registered with a General Practitioner (GP) practice contributing to the Secure Anonymised Information Linkage (SAIL) Databank.

### Data source

The data source was the SAIL Databank, a repository of anonymised patient records for the population of Wales, representing approximately 5 million individuals between January 2000 and December 2021 [31]. Data is collected from approximately 84% of all GP practices in the country. The development, database structure, policies and requisite procedures governing the use of SAIL data have been detailed previously [32–34]. Approval was granted by the Information Governance Review Panel (IGRP) in February 2022 (ref 1371) for a study on SCH and clinical outcomes to be performed using SAIL datasets. The primary care, hospital, outpatient, emergency department, death, demographic and test result datasets (S1 Appendix) were first accessed on 14 March 2022.

### Study cohort

Cohort entry was defined by the presence of one or more of: SCH Read v2 codes, SCH International Classification of Diseases version 10 (ICD-10) codes or test results indicative of SCH within the study timespan. For test results, the respective lab reference ranges were used as recorded to identify the upper limit of TSH and normal levels of T4 (S2 Appendix). Subjects could enter the cohort at any point between 1 January 2000 and 31 December 2021 (Fig 1). Exit from the study occurred at the earliest of: (i) death, (ii) censoring due to the end of GP registration or emigration from Wales or (iii) the end of the study period.

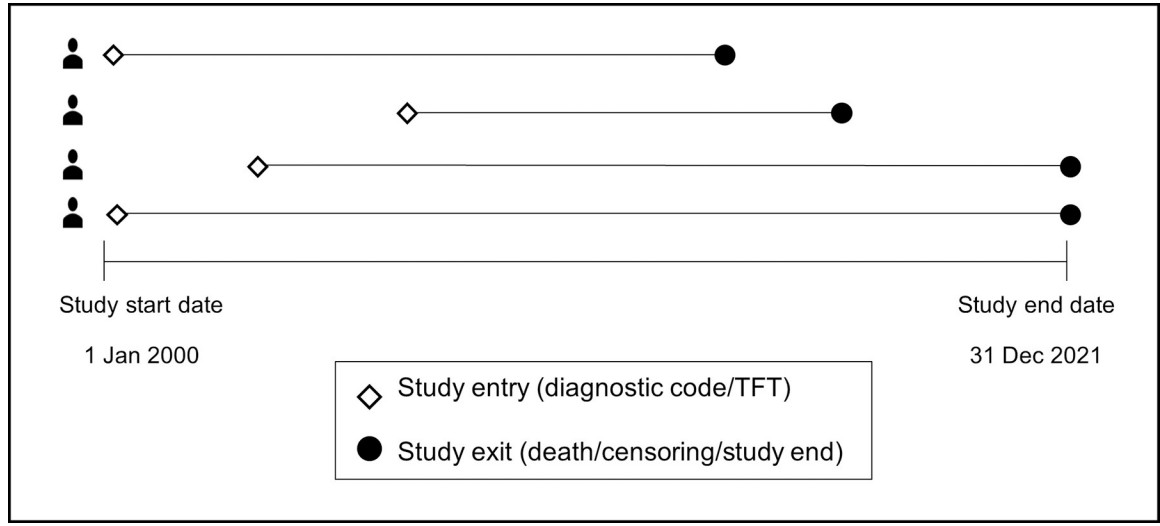

**Fig 1. Study design and examples of possible patient pathways.**

## Eligibility criteria

To be included in the study, patients must have had a recorded SCH diagnostic code or test result indicative of SCH. They must also have been registered with a GP for at least 12 months before the index SCH code or test result.

The following groups were excluded: women with recorded pregnancies within 12 months of the index SCH code or test, patients under 18 years old, those that had ever been identified as having overt hypothyroidism, those given prescriptions for thyroid-altering medications (amiodarone, lithium and antithyroid drugs) or thyroid hormone replacement (more than 30 days before the index code or test) and those with histories of radioiodine and thyroidectomy, such that there were no other indications for the use of levothyroxine.

## Study variables

Baseline characteristics were assessed on the index diagnostic code or test date, which was set as the 'date of identification'. Patients were further classified into two mutually exclusive groups based on whether they received prescriptions for thyroid hormone replacement at any point after the date of identification ('treated') or did not receive any treatment ('untreated').

Sex was recorded as either male or female. Age was calculated from demographic records as the difference in years between the week of birth date and date of identification. This variable was also categorised into bands spanning ten years each. Deprivation scores were enumerated using the 2019 version of the Welsh Index of Multiple Deprivation (WIMD 2019) and assigned based on the recorded home postcode on the date of identification [35, 36]. Length of follow-up was calculated as the difference between the study start date and the exit date or study end date.

Additional SCH-related characteristics such as TSH levels, hypothyroidism codes, and pre-scribed medications were derived from the primary care or test result datasets. The presence of ≥1 thyroid hormone prescription was used as a proxy indicator for treatment status. The number of TFTs performed before and after initiating levothyroxine was also calculated for treated patients. This was done to gauge the frequency with which the NICE treatment

guideline–a repeat abnormal TSH result of $\geq$10 mIU/L after three months before commencing treatment–was followed.

Where available, thyroid peroxidase (TPO) antibody measurements were extracted from the test results dataset and compared to normal ranges. All references to TFT results in this text required that TSH and FT4 results share the same specimen collection date. Hence, 'normal TFT result(s)' refers to TSH and FT4 having both been within their respective reference ranges, whereas 'abnormal TFT result(s)' represents the elevated TSH and normal FT4 characteristic of SCH. Rather than selecting fixed study thresholds for thyroid hormones and antibodies, the reported lab reference ranges were used to align our identification of SCH with the results clinicians would have received.

## Statistical analy5sis

Descriptive statistics were used to summarise baseline demographic data and clinical characteristics. For categorical variables, counts and percentages were used; means and standard deviations were employed for continuous variables.

Estimates of annual cumulative incidence were obtained for each year between 2000 and 2021; this was calculated as the number of newly identified SCH cases between 1 January and 31 December divided by the total number of GP-registered individuals as of 1 July (mid-year population) and multiplied by 100,000. The mid-2011 Welsh population was used to obtain age- and sex-standardised estimates.

The frequencies of normal and abnormal TFT results per patient were classified as prior to or later than the date of identification. We reported the total number of TFTs in the wider GP-registered population to explore if there was a relationship to the number of tests ordered in the same year for the study population. For the former, the study eligibility criteria were applied to all patients who had recorded TFT results between 2000 and 2021 –the number of tests per year was then calculated, irrespective of whether SCH was detected.

The frequency of levothyroxine prescriptions each year and the time between the date of identification and the first prescription were also calculated.

Structured query language (SQL DB2) was used to retrieve and interrogate the SAIL datasets via the SAIL Gateway [37].

## Ethical approval

The project was approved in writing by the SAIL Databank IGRP (ref 1371) and by the School of Medicine Ethics Committee, acting on behalf of the University of St Andrews Teaching and Research Ethics Committee (UTREC) (MD16055) in March 2022. As the study data are de-identified, authors had no access to disclosive information and consent from individual patients was not required.

The results are reported using the REporting of studies Conducted using Observational Routinely-collected Data (RECORD) guidelines [38].

## Results

Between January 2000 and December 2021, over 5.6 million individuals were identified in the demographic register, 304,148 of whom had recorded diagnostic codes or tests indicative of SCH. After applying the eligibility criteria resulting in the exclusion of 104,628 patients, 199,520 individuals were identified as incident SCH cases over the study period (Fig 2). The proportion of untreated patients (n = 152,416; 76.4%) was more than three times that of those who received levothyroxine over the study period (n = 47,104; 23.6%).

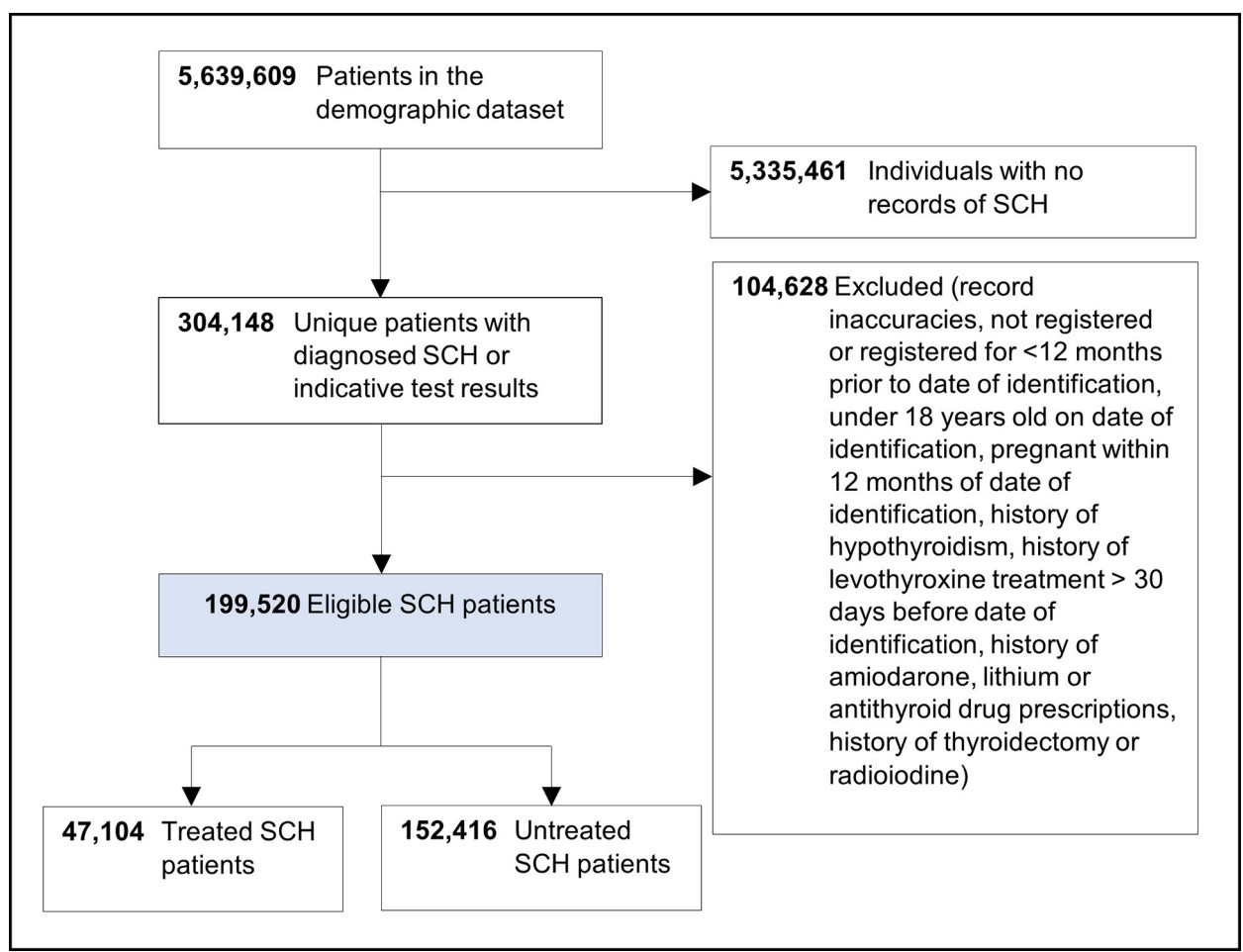

**Fig 2. Study flowchart illustrating the selection of eligible SCH patients using SAIL data.**

### Patient characteristics

The baseline characteristics of the study population are shown in Table 1. There were more female than male patients in the entire cohort (63.8% vs 36.2%). The mean age of study participants was 57.8 years, with a standard deviation of 17.55 years. The highest patient numbers were observed in the 60–69 age band (n = 39,486; 19.8%). There were more untreated than treated patients across all age bands.

The total length of follow-up for the study population was 1,286,883 person-years, with a median duration of 5.75 person-years (IQR 2.65–9.65). 24.6% of the study population died before the end of the study, as opposed to those censored (n = 7,715; 3.9%) because they moved away from Wales or switched to non-SAIL GP practices. Most patients had records running to the end of the study period, 31 December 2021 (n = 142,687; 71.5%).

There was considerable overlap between categories based on the means of identification from the EHR. 99.2% of patients had recorded TFT results on the date of identification; only 149 patients had a combination of all three criteria (Table 1).

**Table 1. Patient characteristics classified by treatment status at baseline and the end of the study period (2000–2021) and the methods used to identify patients using EHR.**

| | | Study population | | |
| --- | --- | --- | --- | --- |
| | | Total (%)[a] | Treated (%)[b] | Untreated (%)[b] |
| | N | 199,520 | 47,104 (23.6) | 152,416 (76.4) |
| **Sex** | | | | |
| | Male | 72,175 (36.2) | 12,169 (16.9) | 60,006 (83.1) |
| | Female | 127,345 (63.8) | 34,935 (27.4) | 92,410 (72.6) |
| **Age (years), mean [SD]** | | 57.8 [17.55] | 54.7 [16.79] | 59.5 [17.71] |
| **Age bands (years)** | | | | |
| | 18–29 | 16,949 (8.5) | 3,529 (20.8) | 13,420 (79.2) |
| | 30–39 | 15,478 (7.8) | 4,583 (29.6) | 10,895 (70.4) |
| | 40–49 | 25,667 (12.9) | 7,878 (30.7) | 17,789 (69.3) |
| | 50–59 | 34,905 (17.5) | 9,696 (27.8) | 25,209 (72.2) |
| | 60–69 | 39,486 (19.8) | 9,175 (23.2) | 30,311 (76.8) |
| | 70–79 | 35,657 (17.9) | 7,263 (20.4) | 28,394 (79.6) |
| | 80–89 | 24,701 (12.4) | 4,130 (16.7) | 20,571 (83.3) |
| | 90–99 | 6,522 (3.3) | 834 (12.8) | 5,688 (87.2) |
| | 100+ | 155 (0.1) | 16 (10.3) | 139 (89.7) |
| **Deprivation WIMD 2019** | | | | |
| | Most deprived | 39,360 (19.7) | 10,022 (25.5) | 29,338 (74.5) |
| | Next most deprived | 42,610 (21.4) | 10,196 (23.9) | 32,414 (76.1) |
| | Middle deprivation | 40,771 (20.4) | 9,588 (23.5) | 31,183 (76.5) |
| | Next least deprived | 36,806 (18.4) | 7,949 (21.6) | 28,857 (78.4) |
| | Least deprived | 32,355 (16.2) | 7,911 (24.5) | 24,444 (75.5) |
| | Missing | 7,618 (3.8) | 1,438 (18.9) | 6,180 (81.1) |
| **Identification of SCH in EHR[c]** | | | | |
| | Read code | 8,195 (4.1) | 5,071 (61.9) | 3,124 (38.1) |
| | ICD-10 code | 1,097 (0.5) | 381 (34.7) | 716 (65.3) |
| | TFT | 197,833 (99.2) | 46,381 (23.4) | 151,452 (76.6) |
| | Read + ICD-10 + TFT | 149 (0.1) | 93 (62.4) | 56 (37.6) |
| **Length of follow-up (person-years)** | | | | |
| | Total | 1,286,883 | | |
| | Median (IQR) | 5.75 (2.65–9.65) | | |
| **Study exit** | | | | |
| | Study ended | 142,687 (71.5) | 36,047 (25.3) | 106,640 (74.7) |
| | Censored[d] | 7,715 (3.9) | 1,506 (19.5) | 6,209 (80.5) |
| | Died | 49,118 (24.6) | 9,551 (19.4) | 39,567 (80.6) |

[a] Percentage of all eligible patients (n = 199,520);

[b] Percentage of the total in the respective category–row percentage;

[c] These groups add up to more than 100% due to overlap;

[d] Due to the end of GP registration or emigration from Wales.

Abbreviations: *SCH*, Subclinical Hypothyroidism; *SD*, Standard Deviation; *WIMD*, Welsh Index of Multiple Deprivation; *ICD*, International Classification of Diseases

## Annual cumulative incidence

The annual cumulative incidence of SCH was irregular over the study period. The number of new cases was highest at a single historical point: approximately 502 cases per 100,000 people in 2012, following a marked decline in 2010 (297 cases per 100,000 people). Between 2015 and

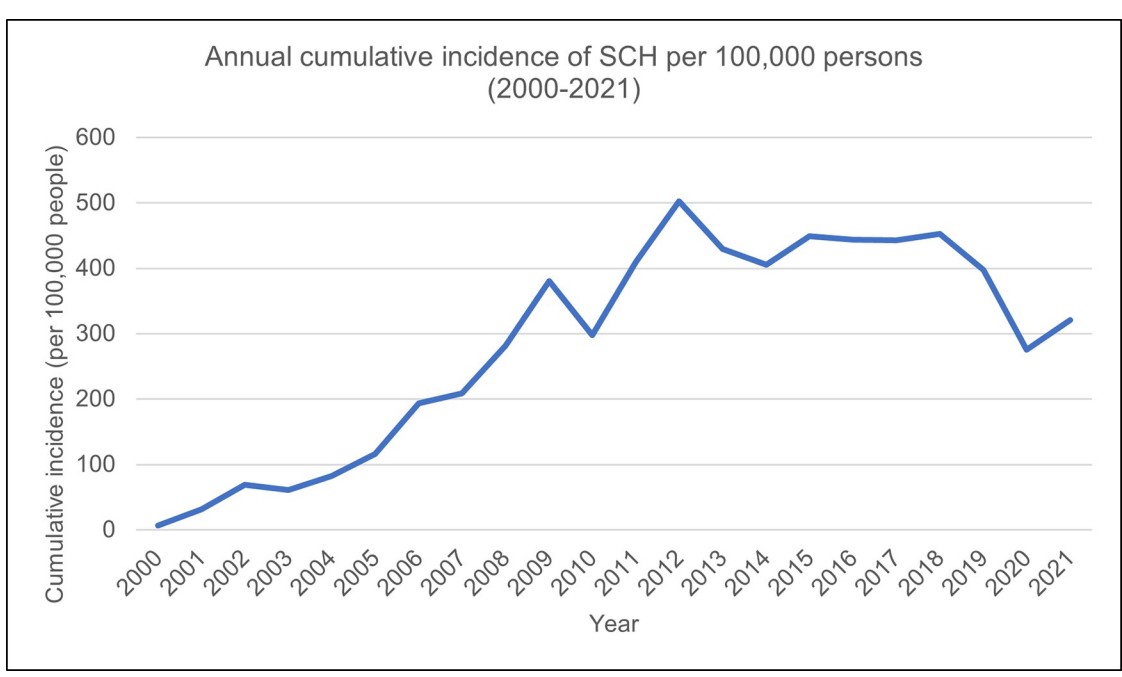

**Fig 3. Trend of the annual cumulative incidence of SCH during the study period (2000–2021).**

2018, the cumulative incidence of SCH was steady but fell again until 2020 (275 cases per 100,000 people) (Fig 3).

When stratified by age and sex, the cumulative incidence charts followed a similar trajectory, though it was noted that the number of female patients was higher for both measures (S3 Appendix).

### Testing and treatment patterns

The annual overall number of TFTs ordered for SCH patients in this study was 1,046 in 2000, and because more patients were identified as having SCH over the study period, the totals were highest in 2018 (n = 123,448) (Fig 4). There was a drop in 2020 (n = 88,000) compared to the previous year (n = 121,448), coinciding with the COVID-19 pandemic. The plotted graph for tests ordered for the wider GP-registered population appears similar to that for study participants but has corresponding peaks and troughs of a larger magnitude (Fig 4).

On the date of identification, 92.5% (n = 184,484) of the study population had TSH values between the upper limit of normal and 10 mIU/L. Of this group, 39,324 (21.3%) received prescriptions for levothyroxine during the study. However, the split between treated and untreated patients was reversed for patients with TSH levels higher than 10 mIU/L (Table 2).

In total, more than half of patients with severe SCH–TSH levels over 10 mIU/L–on the date of identification were in the treated group (n = 6,891; 53%). There were also more patients with elevated TPO antibodies on the recorded date of the first code or test among the treated (n = 5,428; 59.5%) than the untreated group (n = 3,689; 40.5%), as shown in Table 2.

It was also noted that 6,811 (83.1%) of patients identified using Read v2 codes had mild SCH at the time of identification, but almost two-thirds of these patients subsequently received treatment over the study period (n = 4,140; 60.8%).

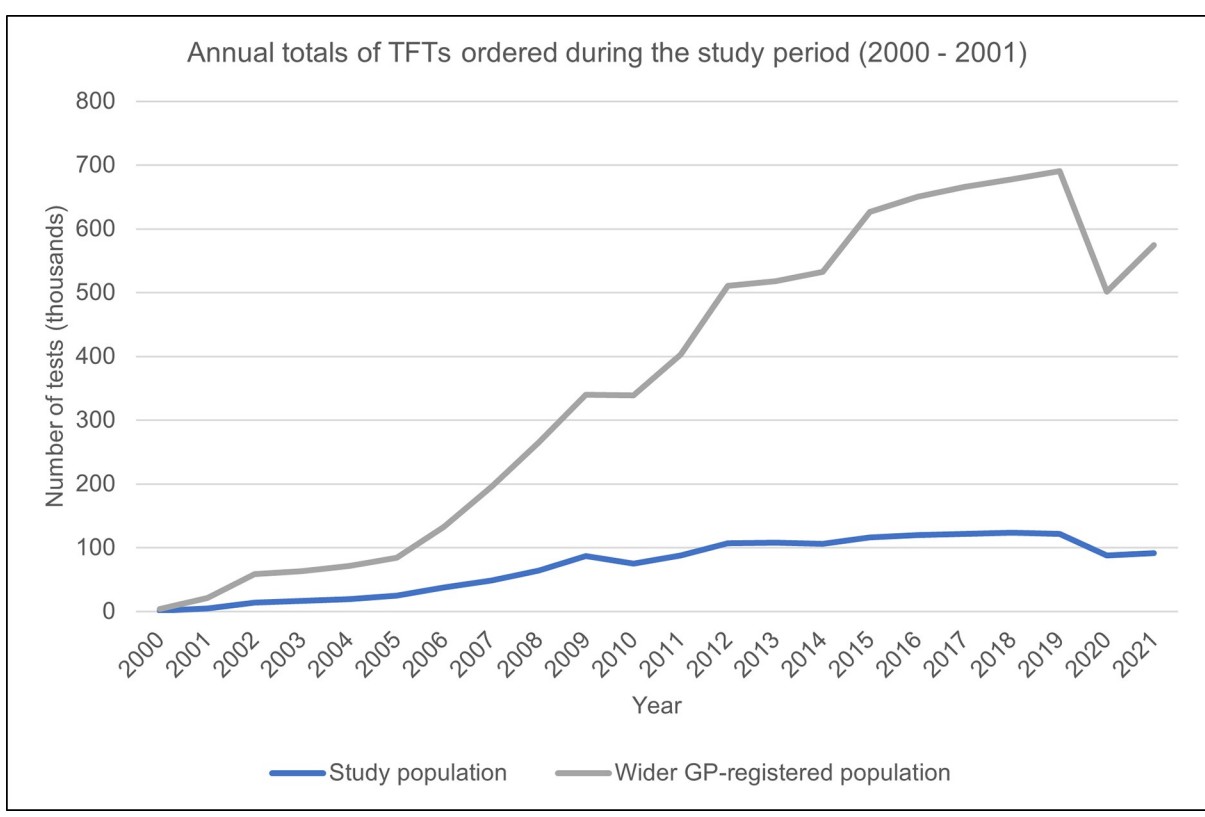

**Fig 4. Annual total number of TFTs ordered for the study participants (n = 199,520) and the wider GP-registered population after applying the study eligibility criteria (n = 1,647,510).**

For the treated group, 19,716 (41.9%) patients received their first levothyroxine prescription after only one test result showing raised TSH and normal T4. (Fig 5).

Most treated patients had between 1 and 5 TFTs performed in the first 12 months (n = 31,840; 67.6%), 24 months (n = 32,801; 69.6%) and 36 months (n = 30,348; 64.4%) after treatment with levothyroxine was initially prescribed (Table 3). The proportion of untreated patients with no recorded follow-up TFTs was larger than that of treated patients over the same three-year period after SCH was identified or treated. Over this duration, more in the treated group (n = 7,399; 15.7%) had more than five monitoring tests compared to the untreated (n = 8,167; 5.4%).

The number of patients on treatment rose gradually, from 52 to 33,337 of the existing SCH cases in 2000 and 2021 respectively (Fig 6). Here, 'initiating treatment' refers to the number of patients who received their first prescription for levothyroxine in that year, as opposed to those that had already commenced treatment since the study start date ('continuing treatment'). Except at the start of the study (2000 to 2002), a smaller proportion of patients were started on treatment throughout (Fig 6 and S4 Appendix).

Under one-fifth of patients in the treated group (n = 7,794; 16.5%) received their first levothyroxine prescription within one month of their index test or code for SCH. In contrast, most treated patients (n = 29,818; 63.3%) got their first prescription more than 12 months after SCH was identified (Table 4).

**Table 2. Recorded TSH and TPO levels on the date of identification, overall and stratified by treatment status.**

| | | Study population | | |
| --- | --- | --- | --- | --- |
| | | Total (%)[a] | Treated (%)[b] | Untreated (%)[b] |
| **TSH on the date of identification** | | | | |
| | Upto 10 mIU/L | 184,484 (92.5) | 39,324 (21.3) | 145,160 (78.7) |
| | 10–20 mIU/L | 10,674 (5.3) | 5,669 (53.1) | 5,005 (46.9) |
| | >20 mIU/L | 2,328 (1.2) | 1,222 (52.5) | 1,106 (47.5) |
| **TPO antibodies on the date of identification** | | | | |
| | Normal | 1,588 (0.8) | 309 (19.5) | 1,279 (80.5) |
| | Elevated | 9,117 (4.6) | 5,428 (59.5) | 3,689 (40.5) |
| | Missing[d] | 188,815 (94.6) | 41,367 (21.9) | 147,448 (78.1) |
| **Patients with SCH Read v2 codes (n = 8,195)[e]** | | | | |
| **TSH on the date of identification** | | | | |
| | Upto 10 mIU/L | 6,811 (83.1) | 4,140 (60.8) | 2,671 (39.2) |
| | 10–20 mIU/L | 452 (5.5) | 363 (80.3) | 89 (19.7) |
| | >20 mIU/L | 53 (0.6) | 48 (90.6) | 5 (9.4) |

[a] Percentage of all eligible patients (n = 199,520);

[b] Percentage of the total in the respective category–row percentage;

[c] These patients did not have TFT results recorded on the index date;

[d] These patients did not have recorded TPO antibody tests on the date of identification;

[e] Not all patients with Read codes also had recorded TFT results.

Abbreviations: *IU*, International Units; *SCH*, Subclinical hypothyroidism; *TSH*, Thyroid Stimulating Hormone; *TPO*, Thyroid Peroxidase

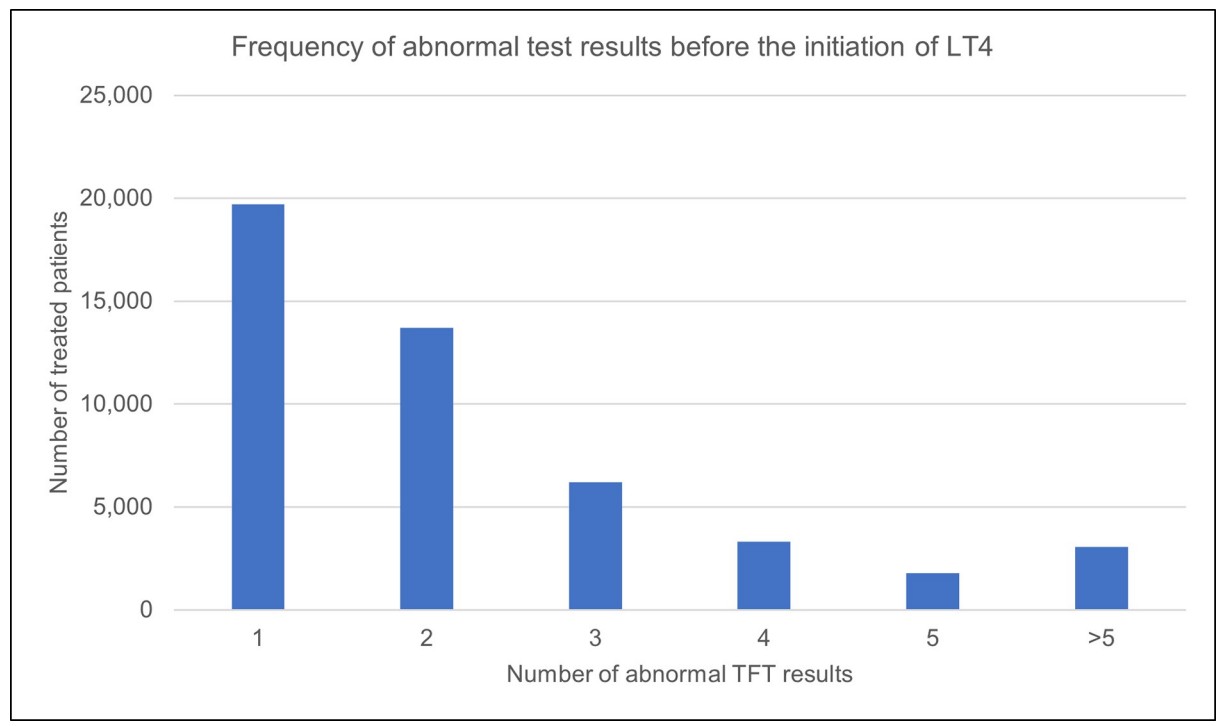

**Fig 5. The frequency of abnormal tests before treatment with levothyroxine was initiated for the treated group (n = 47, 104).**

**Table 3. The number of follow-up thyroid function tests performed in the immediate period (i) after the first prescription, date for treated patients and (ii) after identification, for untreated patients.**

| Number of monitoring TFTs performed | Treated patients (n = 47,104) | Untreated patients (n = 152,416) |
|---|---|---|
| | After the first prescription (%) | After identification (%) |
| **First year** | | |
| **None recorded** | 14,372 (30.5) | 65,797 (43.2) |
| **1–5** | 31,840 (67.6) | 85,622 (56.2) |
| **>5** | 892 (1.9) | 997 (0.7) |
| **First two years** | | |
| **None recorded** | 10,194 (21.6) | 46,650 (30.6) |
| **1–5** | 32,801 (69.6) | 101,807 (66.8) |
| **>5** | 3,747 (8.0) | 3,512 (2.3) |
| **First three years** | | |
| **None recorded** | 8,449 (17.9) | 39,347 (25.8) |
| **1–5** | 30,348 (64.4) | 103,887 (68.2) |
| **>5** | 7,399 (15.7) | 8,167 (5.4) |

## Discussion

Between 2000 and 2021, 199,520 individuals residing in Wales were identified as having SCH. The annual cumulative incidence of SCH was irregular, with a marked drop in 2010 and a prominent peak in 2012 for crude and both age- and sex-standardised estimates. In keeping with other studies on SCH [11, 39], it was found that more females than males were identified

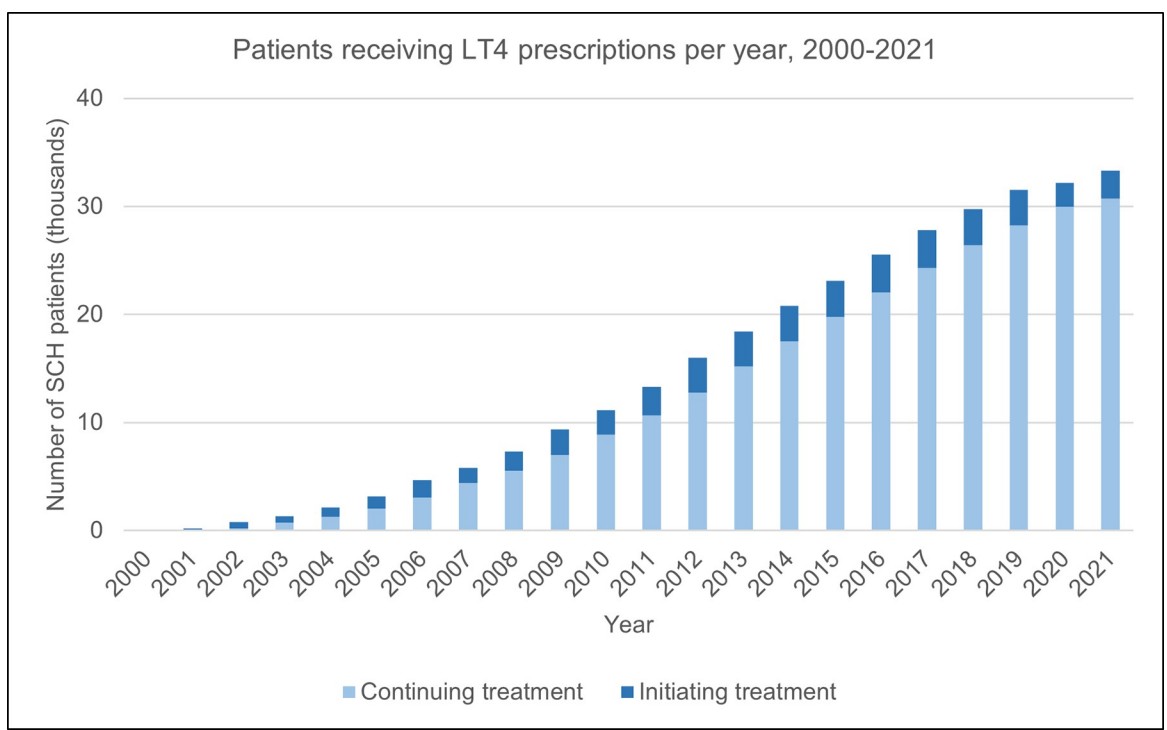

**Fig 6. The number of SCH patients with recorded prescriptions for levothyroxine during the study period.**

**Table 4. The characteristics of the treatment given to SCH patients in Wales, 2000–2021.**

| | | Treated patients (n = 47,104) |
|---|---|---|
| **Time from the date of identification to the first LT4 prescription** | | |
| | ≤1 month | 7,794 (16.5) |
| | 1–6 months | 6,470 (13.7) |
| | 6–12 months | 3,022 (6.4) |
| | > 1 year | 29,818 (63.3) |

Abbreviations: *LT4*, Levothyroxine

with SCH. Less than one-third of the study population received treatment during the study period. Most patients identified using diagnostic codes, specifically Read v2, received levothyroxine over the study period, though an equally large proportion of these patients also had mild SCH on their first test date.

In contrast, 6.5% of the study population had TSH levels higher than 10 mIU/L on their index test results. Levothyroxine was prescribed for 42% of patients in the treated group after a single abnormal test indicative of SCH. The frequency of TFTs for treated patients after their first prescription was higher than tests performed for untreated patients after identifying SCH.

## Annual cumulative incidence

The reason for the peak in annual cumulative incidence in 2012 was not immediately apparent. However, most of the patients in this study were identified using TFT results rather than diagnostic codes. A potential reason for the irregular pattern of annual incident cases, therefore, might be altered clinical decision-making as a result of new guidance; for example, the Royal College of Physicians released a statement on the diagnosis and management of primary hypothyroidism in 2008 [40] and updated the guidance in 2011 [41]. The intervening period corresponds to the marked drop in incident cases of SCH in the study population around 2010, but this conclusion cannot be reached based on a single study.

Notably, when standardised to the mid-2011 population of Wales, estimates of the annual cumulative incidence of SCH were broadly similar across the age and sex categories. However, the line representing patients aged 90 and over was flattened compared to the crude annual cumulative incidence plot, suggesting that the proportion that was 90 years or older in the standard population was smaller than among the study population.

## Testing and treatment

Overtesting for thyroid function has been widely reported in the literature [28, 29, 42–44]. In contrast, our findings show that in the years following the identification of SCH, most of the untreated patients had fewer tests, which aligns with current guidance to perform repeat TFTs annually or biennially for untreated SCH, depending on the presence of features of underlying thyroid disease [25]. Treated patients, who would typically require frequent monitoring tests–every three months until TSH levels normalise and then annually–were found to have had more TFTs in comparison. Annual monitoring would possibly explain why 64% of treated patients had between one and five tests in the first three years after treatment was started.

Data on TPO antibodies on the date of identification were available for only a small fraction of patients in the cohort. However, this corresponds to the NICE recommendation to consider antibody testing for elevated TSH levels and avoid repeating these specific tests [25]. TPO antibody assays predating these NICE guidelines would also likely have been infrequent.

The plotted rise in the number of TFTs ordered for study participants can be explained by the increasing number of identified SCH cases over the study period. On the other hand, the marked drop observed around 2020 may be attributed to the COVID-19 pandemic, during which mandatory lockdowns were implemented. It is, however, impossible to completely rule out overtesting in this study because inappropriate TFT ordering accompanied by a higher number of patients receiving tests would, unsurprisingly, present a false picture of a steady average among patients with SCH.

Over 90% of the study population had mild SCH–TSH levels between the upper limit of normal and the 10mIU/L threshold–on the date they were identified. This number is higher than that reported from the Colorado Thyroid Disease Prevalence study, 74% [11]. However, the participants in the Colorado study were the attendees of a statewide health fair, representing 25,862 individuals. Unlike this longitudinal study based on SCH cases over 20 years, the fair facilitated a cross-sectional survey. As such, a more extended series of surveys might have approximated our findings more closely.

Patients with a recorded diagnosis were more likely to receive treatment than those identified through test results. Of the latter, less than a third were in the treated group, suggesting that GPs may have been more likely to disregard test results indicative of SCH if, for instance, the TFTs were not directly relevant to their plans for the clinical management of presenting symptoms. In such a scenario, the SCH diagnosis would not be recorded, decreasing the probability of the patient receiving a prescription for levothyroxine. Crucially, the study findings also indicated that the presence of a diagnostic code in the EHR was not an indicator of the severity of SCH–as determined using the 10 mIU/L threshold for TSH. Their index test results showed that most patients with clinically recorded diagnoses had mild SCH in the first instance.

More of the patients who had severe SCH on the index date received treatment during the study, as would be expected. Several treatment guidelines around the world similarly recommend the initiation of levothyroxine for cases with TSH levels above 10mIU/L, including those from the American Thyroid Association [45, 46], European Thyroid Association [47] and Brazilian Society of Endocrinology and Metabolism [48]. A crucial caveat to this guidance is that patient age must be considered, given the physiological increases in TSH levels with age [20, 23]. It is frequently stated that a higher treatment threshold should be applied for older patients, particularly for mild SCH [18, 49, 50]. Even so, a multinational survey on treatment practices for SCH by Razvi et al. [51] found wide variability in the implementation of such recommendations. Potential reasons for the failure to treat cases of severe SCH, as observed for 3.1% of our study population, include patient age, transient increases in TSH or measurement errors [2], which would be lower on subsequent tests.

Levothyroxine is currently the third most frequently prescribed drug by GPs in Wales [52–54]. Previous studies have also described 'overtreatment'–the tendency for clinicians to initiate treatment even when it is not necessarily required and would not benefit the patient [30, 55]. The finding that 42% of all the patients in the treated group had levothyroxine initiated based on a single abnormal test result indicative of SCH may be related to overtreatment. This proportion was notably higher than the number of patients with severe SCH and those who tested positive for TPO antibodies–according to current guidelines, these two groups would have been deemed eligible for immediate treatment. Furthermore, the number of patients who had severe SCH on the date of identification was lower than those who received their first prescription within one month of that date. All these findings point to the potential overuse of levothyroxine over the study period.

### Strengths and limitations

The strengths of this study included the use of multiple SAIL datasets to identify patients identified as having SCH and to assess their eligibility for this study; the scale and coverage of the data used, which is almost the whole population of Wales spanning at least two decades, as well as the characterisation of different aspects of SCH and SCH patients to provide a comprehensive description of the disorder and how it was managed clinically.

This study is subject to some limitations. Foremost are the inherent challenges of identifying SCH in EHR: missing data, coding errors and inconsistencies in patient records influenced the inclusion of patients in the study. It was observed that the test results dataset, in particular, was plagued by these issues due to the variety of ways in which test names, codes and values were recorded. Even so, this problem was mitigated by including the primary care and inpatient datasets to identify patients who had been clinically diagnosed with the disorder. SAIL Databank is also recognised as a high-quality primary care data source with a high level of coverage in Wales (84%) [56]. Also, it was not possible to identify the ethnicity of eligible patients as this information was not available in the provisioned datasets.

Related to these points, most of the study population was identified in the EHR using their TFT results. However, these are not error-proof; it has been reported previously that due to the use of population reference ranges, an individual may have a physiologically abnormal thyroid function test (TFT) result that matches what is otherwise considered normal range [1, 57].

It is also essential to note that for the evaluation of treatment, only prescribing data–not dispensing data–were available. It was, therefore, impossible to ascertain whether the levothyroxine prescriptions given were filled (adherence) and the medication taken by the patient as instructed (compliance). This affected the interpretation of treatment duration, as it would otherwise have been possible to explore LT4 use by accounting for the amount of medication given and, possibly, to distinguish between treated patients by dosages and drug formulations. However, it can reasonably be expected that most patients who received prescriptions took their medication; hence, we used prescriptions as a proxy for treatment status.

A key challenge in estimating the true prevalence of SCH is the lack of agreement concerning TSH reference ranges [9]. Diagnosis relies entirely on lab results; therefore, variations in the upper limit of normal for TSH directly influence patient numbers. However, separate constraints arise from using EHR only to measure prevalence. Chief among these is the likelihood of underestimating the number of existing cases because of the complex interplay of factors that affect the decision to seek clinical care [58]. This selection bias arises because patients with more severe symptoms are more likely to visit their GP and have a recorded visit in the EHR [59, 60]. Another factor is that the restriction of study start and stop dates can mimic a closed cohort in which the 'existing patient' count starts at zero. We did not use the study population for prevalence estimates for these reasons.

Finally, the most recently reported SAIL coverage of GP practices in Wales is approximately 84% [56]; therefore, it cannot be assumed that the denominators used to estimate the annual cumulative incidence are directly equal to the respective actual population counts. This challenge was mitigated by performing standardisation of the estimated cumulative incidence, though the plotted graphs were essentially unchanged.

### Conclusion

This descriptive study on SCH in Wales shows an uneven rise in the overall number of patients, TFT and levothyroxine use between 2000 and 2021, with the highest annual cumulative incidence in 2012 at 502 cases per 100,000 people.

Compared to patients who only had test results indicative of SCH, those with clinically recorded diagnoses were less likely to meet the 10mIU/L TSH cutoff for severe SCH but were also more likely to be offered treatment. In contrast, patients with mild SCH on their index test were less likely to receive levothyroxine. The clinical management of SCH was inconsistent with the recommendation to consider treatment only if a repeat test reveals TSH levels higher than 10mIU/L, considering that over a third of treated cases had only one prior abnormal test result. However, per the current NICE annual monitoring guidance, TFTs were ordered more frequently for treated than untreated patients.

Our findings show that more robust guidelines are needed to ensure the appropriate clinical management of patients with SCH. Given the potential for a continued rise in patient numbers and conflicting evidence on the impact of SCH on patients' long-term health, more research is needed to inform strategies to improve the use of TFTs and levothyroxine for managing this thyroid disorder.

## Supporting information

**S1 Appendix. Description of SAIL datasets.**
(DOCX)

**S2 Appendix. Case definition (SCH patients).**
(DOCX)

**S3 Appendix. Age- and sex-stratified annual cumulative incidence according to the mid-2011 Welsh census data.**
(DOCX)

**S4 Appendix. Levothyroxine use during the study period.**
(DOCX)

## Author Contributions

**Conceptualization:** Brenda S. Bauer, Amaya Azcoaga-Lorenzo, Utkarsh Agrawal, Adeniyi Francis Fagbamigbe, Colin McCowan.

**Data curation:** Brenda S. Bauer.

**Formal analysis:** Brenda S. Bauer.

**Funding acquisition:** Colin McCowan.

**Investigation:** Brenda S. Bauer.

**Methodology:** Brenda S. Bauer.

**Project administration:** Brenda S. Bauer.

**Resources:** Colin McCowan.

**Supervision:** Amaya Azcoaga-Lorenzo, Utkarsh Agrawal, Colin McCowan.

**Validation:** Amaya Azcoaga-Lorenzo, Utkarsh Agrawal, Adeniyi Francis Fagbamigbe, Colin McCowan.

**Visualization:** Brenda S. Bauer.

**Writing – original draft:** Brenda S. Bauer.

**Writing – review & editing:** Brenda S. Bauer, Amaya Azcoaga-Lorenzo, Utkarsh Agrawal, Adeniyi Francis Fagbamigbe, Colin McCowan.

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
