## [Decision Letter · Decision Letter 0]

1 Dec 2023

PONE-D-23-32464Subclinical hypothyroidism in Wales from 2000 to 2021: a descriptive cohort study based on electronic health recordsPLOS ONE

Dear Dr. Bauer,

Thank you for submitting your manuscript to PLOS ONE. After careful consideration, we feel that it has merit but does not fully meet PLOS ONE’s publication criteria as it currently stands. Therefore, we invite you to submit a revised version of the manuscript that addresses the points raised during the review process.

We look forward to receiving your revised manuscript.

Kind regards,

Fateen Ata, MD

Academic Editor

PLOS ONE

Additional Editor Comments:

We have completed the peer-review process, and I'm writing to inform you that your submission has been evaluated as requiring major revisions.

The reviewers have provided constructive feedback and identified key areas that need substantial revision. Their comments are aimed at enhancing the scientific rigor, clarity, and impact of your work. We believe your research will benefit from a revision, and we encourage you to revise the manuscript per the comments provided.

Please find the reviewers' comments attached.

Reviewers' comments:

Reviewer's Responses to Questions

**Comments to the Author**

1. Is the manuscript technically sound, and do the data support the conclusions?

Reviewer #1: Yes

Reviewer #2: Yes

2. Has the statistical analysis been performed appropriately and rigorously? 

Reviewer #1: I Don't Know

Reviewer #2: Yes

3. Have the authors made all data underlying the findings in their manuscript fully available?

Reviewer #1: Yes

Reviewer #2: Yes

4. Is the manuscript presented in an intelligible fashion and written in standard English?

Reviewer #1: Yes

Reviewer #2: Yes

5. Review Comments to the Author

Reviewer #1: The demographic and clinical characteristics of patients diagnosed with subclinical hypothyroidism in Wales between 2000 and 2021 are described in this study. Furthermore, the annual cumulative incidence during this period and the testing and treatment patterns linked to the disorders were investigated.

Although the sample size of the study was relatively large and the results obtained in this study provided considerable significance for the assessment of subclinical hypothyroidism, the authors should address the following requirements.

1. Some of the tables need to be changed as bar graphs or pie charts.

2. There was too much redundancy in the conclusion section. The authors need to concisely summarize or itemize the results.

Reviewer #2: In this article Brenda S Bauer and collegues report an discreptive study about the characterisation and the medication of SCH patients which covers almost the whole population of Wales spanning two decades, this study is of some interest,which indicates that more robust guidelines and recommendations are needed to ensure the consistent and appropriate clinical management of patients identified as having SCH.

6. PLOS authors have the option to publish the peer review history of their article (what does this mean?). If published, this will include your full peer review and any attached files.

Reviewer #1: **Yes: **Ken-ichi Aihara

Reviewer #2: No

---

## [Author Response · Author response to Decision Letter 0]

13 Jan 2024

Reviewer 1:

We thank Reviewer 1 for their feedback on the manuscript. Because there are already five figures included in our original submission, we have limited scope to create additional figures. However, we have converted Table 3, which showed the frequency of abnormal tests before levothyroxine was initiated, to a bar graph – this is now included as Figure 5.

The original length of the Conclusion section was 227 words – this has now been shortened to 210 words and rearranged to improve flow. It now reads:

“This descriptive study on SCH in Wales shows an uneven rise in the overall number of patients, TFT and levothyroxine use between 2000 and 2021, with the highest annual cumulative incidence in 2012 at 502 cases per 100,000 people. 

Compared to patients who only had test results indicative of SCH, those with clinically recorded diagnoses were less likely to meet the 10mIU/L TSH cutoff for severe SCH but were also more likely to be offered treatment. In contrast, patients with mild SCH on their index test were less likely to receive levothyroxine. The clinical management of SCH was inconsistent with the recommendation to consider treatment only if a repeat test reveals TSH levels higher than 10mIU/L, considering that over a third of treated cases had only one prior abnormal test result. However, per the current NICE annual monitoring guidance, TFTs were ordered more frequently for treated than untreated patients. 

Our findings show that more robust guidelines are needed to ensure the appropriate clinical management of patients with SCH. Given the potential for a continued rise in patient numbers and conflicting evidence on the impact of SCH on patients’ long-term health, more research is needed to inform strategies to improve the use of TFTs and levothyroxine for managing this thyroid disorder.”

Reviewer 2: 

We thank Reviewer 2 for their comments on the manuscript.

---

## [Decision Letter · Decision Letter 1]

1 Feb 2024

Subclinical hypothyroidism in Wales from 2000 to 2021: a descriptive cohort study based on electronic health records

PONE-D-23-32464R1

Dear Dr. Bauer,

We’re pleased to inform you that your manuscript has been judged scientifically suitable for publication and will be formally accepted for publication once it meets all outstanding technical requirements.

Kind regards,

Fateen Ata, MD

Academic Editor

PLOS ONE

Additional Editor Comments (optional):

All comments have been addressed and manuscript is now suitable for publication.

Reviewers' comments:

Reviewer's Responses to Questions

**Comments to the Author**

1. If the authors have adequately addressed your comments raised in a previous round of review and you feel that this manuscript is now acceptable for publication, you may indicate that here to bypass the “Comments to the Author” section, enter your conflict of interest statement in the “Confidential to Editor” section, and submit your "Accept" recommendation.

Reviewer #1: All comments have been addressed

2. Is the manuscript technically sound, and do the data support the conclusions?

Reviewer #1: Yes

3. Has the statistical analysis been performed appropriately and rigorously? 

Reviewer #1: Yes

4. Have the authors made all data underlying the findings in their manuscript fully available?

Reviewer #1: No

5. Is the manuscript presented in an intelligible fashion and written in standard English?

Reviewer #1: Yes

6. Review Comments to the Author

Reviewer #1: The authors have revised the manuscript after the first round review process. I have no further comment.

7. PLOS authors have the option to publish the peer review history of their article (what does this mean?). If published, this will include your full peer review and any attached files.

Reviewer #1: **Yes: **Ken-ichi Aihara

---

## [Editor Report · Acceptance letter]

9 May 2024

PONE-D-23-32464R1 

PLOS ONE

Dear Dr. Bauer, 

I'm pleased to inform you that your manuscript has been deemed suitable for publication in PLOS ONE. Congratulations! Your manuscript is now being handed over to our production team.

Kind regards, 

on behalf of

Dr. Fateen Ata 

Academic Editor

PLOS ONE